

# Momentum-space and real-space Berry curvatures in Mn₃Sn

Xiaokang Li[1], Liangcai Xu[1], Huakun Zuo[1],
Alaska Subedi[2,3], Zengwei Zhu[1*], Kamran Behnia[4†]

**1** Wuhan National High Magnetic Field Center and School of Physics,
Huazhong University of Science and Technology, Wuhan 430074, China
**2** Centre de Physique Théorique, École Polytechnique,
CNRS, Université Paris-Saclay, 91128 Palaiseau, France
**3** Collège de France, 11 place Marcelin Berthelot, 75005 Paris, France
**4** Laboratoire de Physique et d'Étude des Matériaux (UPMC-CNRS),
ESPCI Paris, PSL Research University 75005 Paris, France

★ zengwei.zhu@hust.edu.cn
† kamran.behnia@espci.fr

## Abstract

Mn₃X (X= Sn, Ge) are noncollinear antiferromagnets hosting a large anomalous Hall effect (AHE). Weyl nodes in the electronic dispersions are believed to cause this AHE, but their locus in the momentum space is yet to be pinned down. We present a detailed study of the Hall conductivity tensor and magnetization in Mn₃Sn crystals and find that in the presence of a moderate magnetic field, spin texture sets the orientation of the $k$-space Berry curvature with no detectable in-plane anisotropy due to the $Z_6$ symmetry of the underlying lattice. We quantify the energy cost of domain nucleation and show that the multidomain regime is restricted to a narrow field window. Comparing the field dependence of AHE and magnetization, we find that there is a distinct component in the AHE which does not scale with magnetization when the domain walls are erected. This so-called 'topological' Hall effect provides indirect evidence for a non-coplanar spin components and real-space Berry curvature in domain walls.


---

# 1  Introduction

The $Mn_3X$ (X=Sn, Ge) family of compounds crystallizing in the $DO_{19}$ hexagonal close-packed Bravais lattice are triangular antiferromagnets with a Néel temperature of around 400 K [1,2]. The recent observation of the anomalous Hall effect (AHE) in these systems [3–5] followed theoretical predictions [6,7] of nonvanishing Berry curvature in a noncollinear yet planar antiferromagnet. The discovery was followed by the detection of anomalous Nernst [8,9] and anomalous Righi-Leduc [9] effects, the thermoelectric and thermal counterparts of the AHE, respectively. The latter observations confirmed that the zero-field transverse anomalous currents are due to the Fermi-surface quasiparticles, as argued by Haldane [10]. Several *ab initio* calculations [8,11] have found an anomalous Hall conductivity (AHC) matching what experiments find at low temperatures. However, the precise configuration of spins and the locus of the Weyl nodes in the *k*-space generating the Berry curvature that cause these phenomena are still subject to debate.

The Hall resistivity of $Mn_3Sn$ has a peculiar profile (see Fig. 1). Recognizably different from the AHE signal resolved in ordinary ferromganets like body-centred cubic iron [9,12,13] or cobalt [14], it is also quite distinct from the much-studied spiral helimagnet MnSi [15,16]. In contrast to these cases, in $Mn_3Sn$, $\rho_{ij}$ presents a hysteretic jump dwarfing the slope caused by the ordinary Hall resistivity. The hysteresis has a shape unlike the sigmoid commonly seen in ferromagnets [17]. Finally, the asymmetry of this loop contrasts with the symmetric hysteresis of the quantum AHE observed in magnetic two-dimensional topological insulators [18,19].

In this paper, we show that the peculiarity of this hysteresis loop resides in the existence of a threshold field $B_0$ for domain nucleation. Three distinct regimes can be identified. In regime I, below $B_0$, there is a single magnetic domain with an orientation set by the sample history and not by the applied field. In regime II, above $B_0$, multiple domains coexist, and, as the magnetic field increases, the domain favored by it occupies a larger portion of the sample. At sufficiently

higher fields (regime III), the sample becomes single domain again, and the domain orientation is now entirely set by the magnetic field. Monitoring the electric field generated by a rotating in-plane magnetic field in regime III, we find that the finite component of the AHC tensor is set by the orientation of spins and not by the underlying lattice. This observation is backed by theoretical calculations, which find that the single-ion anisotropy is vanishingly small and the Fermi surface is not modified by rotation of spins. It also has implications for the ongoing effort to pin down the source of the Berry curvature in the reciprocal space. The magnitudes of $B_0$ and the jump in magnetization can be used to quantify the energy cost of erecting domain walls. Finally, we compare the field dependences of $\rho_{ij}^A$ and magnetization and find that they start to deviate from each other in the multidomain regime. Such an observation in other systems has been commonly attributed to a 'topological Hall effect' (THE) due to the accumulation of the Berry curvature in the real space [20], which can be caused by a nontrivial magnetic texture, such as a skyrmion lattice in the phase A of MnSi [15]. We suggest that in the multidomain regime, domain walls generate a real-space Berry curvature and an additional contribution on top of the one caused by momentum-space Berry curvature. This would imply a non-coplanar spin component for domain walls, which is yet to be observed by microscopic probes.

## 2 Three distinct regimes in the hysteresis loop

Fig. 1 shows the hysteretic loop of $\rho_{ij}(B)$ at room temperature in a millimetric $Mn_3Sn$ single crystal. When the magnetic field attains a magnitude as low as 0.2 T, $\rho_{ij}$ locks into a finite magnitude and does not show any further evolution except a tiny slope due to the ordinary Hall effect. When the field is swept back to zero, $\rho_{ij}(B)$ remains locked to its magnitude. Only when the field, oriented along the opposite direction, attains a specific amplitude, which we call $B_0$, $\rho_{ij}(B)$ begins to change steeply. Upon further increase, $\rho_{ij}(B)$ saturates to a value opposite in sign but identical in magnitude to its initial value. As seen in the figure, repeating this procedure numerous times with different sweeping rates reproduces the same curve. This is very different from the hysteretic magnetization profile seen in ferromagnets, which has a "sigmoid" shape [17]. On the other hand, it is remarkably similar to the hysteretic loop of magnetization resolved in a ferromagnetic liquid crystal [21]. In the former case, the shape of the hysteretic loop is set by the displacement of domain walls and their pinning by defects. The loop is smooth, and the passage between single-domain and multidomain regimes in its two ends are symmetric [17].

The existence of a finite threshold field for domain nucleation implies that, below this field, tolerating a magnetization opposite to the applied field is less costly in energy than erecting a domain wall. At $B_0$, the two costs become equal and domain reversal starts. Insensitivity to the sweep rate suggests thermodynamic equilibrium during the entire loop. In other words, the time scale of all detectable dynamic phenomena remain faster than our sweeping rates. The boundary between regime I (single-domain) and regime II (multidomain) is sharp, but the boundary between regime II and regime III (field-induced single domain) is fuzzy and, as we will see below, hosts a specific component in $\rho_{ij}(B)$ generated by inhomogeneous magnetization. In regime III, the signal smoothly saturates to its initial magnitude with an opposite sign, indicating an inverted single-domain regime.

## 3 Angle-dependent Hall conductivity

Our angle-dependent study illustrates the difference between the three regimes. In this experiment, electric current was applied along the $z$ axis and the magnetic field rotated in the $xy$

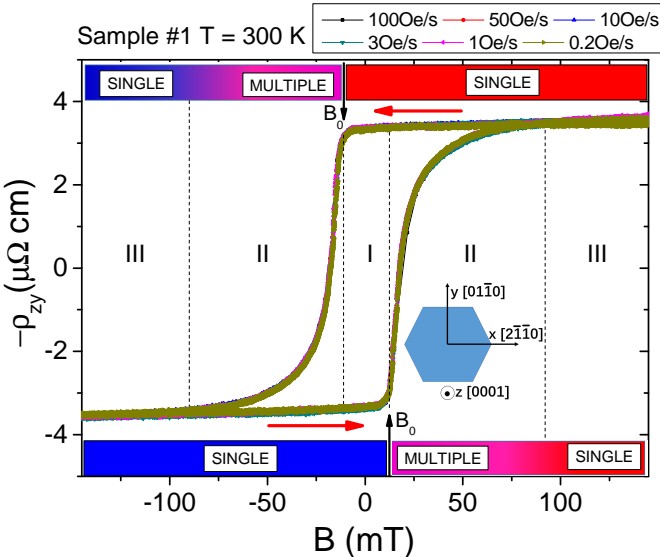

Figure 1: **Hysteretic anomalous Hall effect in Mn$_3$Sn:** $-\rho_{zy}^{A}$ as a function of magnetic field in Mn$_3$Sn has a peculiar shape. Three different regimes can be identified. When the field is lower than $B_0$, marked by a vertical arrow, the spins keep their configuration despite the magnetic field (regime I). When $B > B_0$ (regime II), domains conform to the orientation of the field are induced. At sufficiently high fields, these new domains occupy the bulk of the sample, which becomes single-domain again (regime III). The hysteretic loop is reproducible even when the sweeping rate changes by a factor of 500.

plane. The electric field along different orientations was monitored using multiple electrodes. In this way, we could determine the amplitude and the orientation of the total electric field vector for an arbitrary orientation of magnetic field. We studied both a square sample (see Appendix) and a triangular sample (see Fig. 2), whose shape excluded demagnetization artifacts. The results were similar. In regime I, that is below $B_0$, the electric field was unaffected by rotation. In regime II, strong hysteresis was observed in the angular dependence of the signal. In regime III, each projection of the electric field along the three $x$ and three $y$ axes was found to display almost perfect sinusoidals. Thus, in this regime, the spin texture is easily rotated by a magnetic field. As seen in the bottom panels of Fig. 2, in both regime I and regime III, this is when the system is single domain and the amplitude of the electric field is the same irrespective of orientation.

## 4 Discussion

### 4.1 Single-ion anisotropy

The Hamiltonian relevant to this spin texture, formulated first by by Liu and Balents [22], consists of terms with three distinct energy scales: the Heisenberg exchange, the Dzyaloshinskii-Moriya (DM) interaction and the single-ion anisotropy. In the absence of the latter, U(1) degeneracy is preserved and any in-plane rotation of spins leaves the energy unchanged [22]. Therefore, our experiment implies that this single-ion anisotropy is vanishingly small and what lifts the U(1) degeneracy is the in-plane magnetic field. Therefore, instead of having six domains

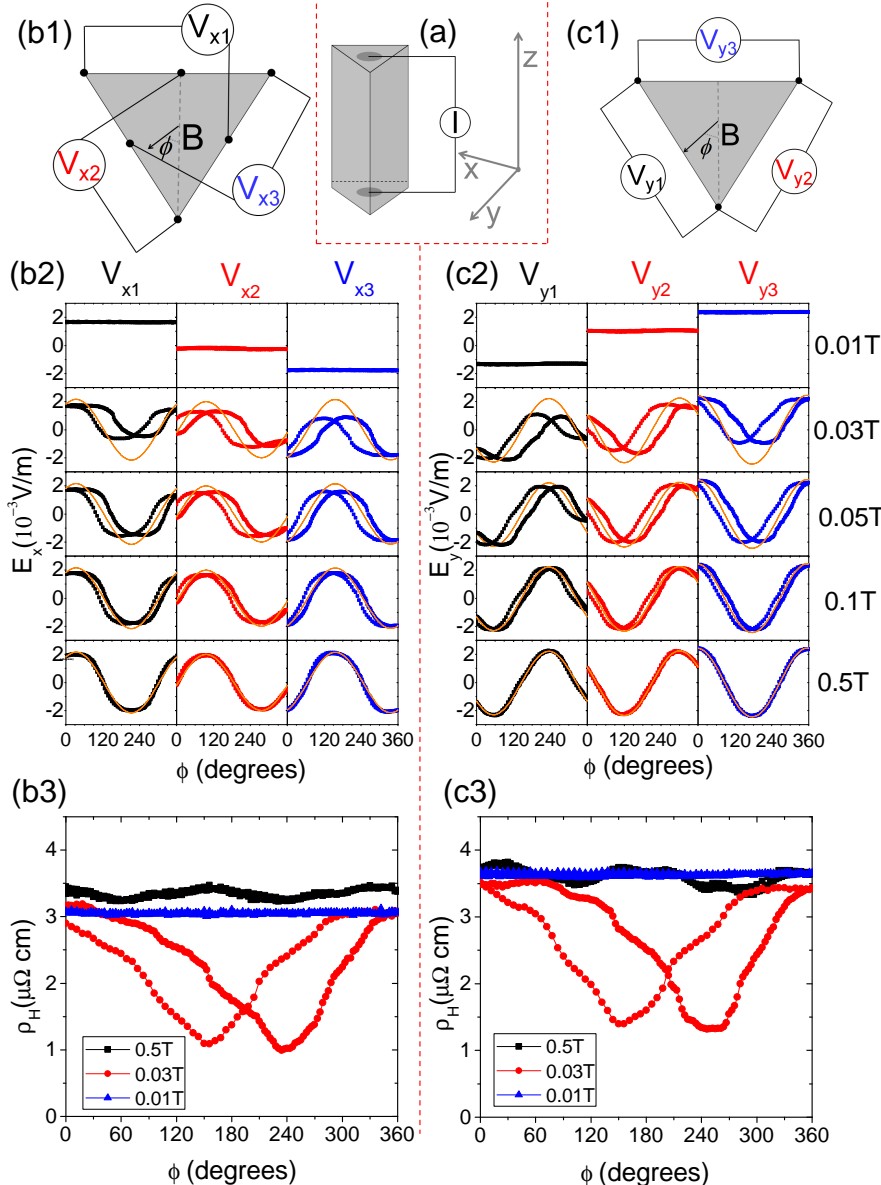

Figure 2: **Angle-dependent Hall resistivity:** a) The current was applied along the $z$ axis of a sample with a triangular cross section and the magnetic field was rotated in the basal plane. Each pair of electrodes monitored the electric field along one of the three equivalent $x$ axes (b1) or the three $y$ axes (c1). Angular variation of the three $E_x$ and the three $E_y$ as a function of the angle between the magnetic field and $x$ axis are shown in (b2) and (c2). At low magnetic field (regime I), the total electric field remains unchanged. At high magnetic field (regime III), the measured electric field becomes sinusoidal (shown as a red line). In the intermediate field range (regime II), the electric field is non-sinusoidal and strongly hysteretic. Panels (b3) and (c3) show $\rho_H = |E|/J$, where $|E|$ represents the magnitude of the total electric field vector extracted from its projections. It is almost the same in regimes I and III. The fluctuations in regime III set an upper bound to any in-plane anisotropy undetectable by this experiment.

(which would have been the case if the degeneracy was lifted by the single-ion anisotropy), we have only two domains set by the orientation of the magnetic field. This is the first outcome of our study. A field as small as 0.5 T easily sets the orientation of the spins by coupling to the in-plane magnetization, which is $1.5 \times 10^{-2} \mu_B$/f.u. at 0.5 T. This corresponds to an energy as small as 0.5 $\mu$eV/f.u. Such an upper boundary is in agreement with the angle-dependent torque magnetometry data reported by Duan and co-workers who quantified the magnetic crystalline anisotropy of the system [24](see section H in Appendix).

## 4.2 Momentum-space Berry curvature

A second consequence is about the in-plane anisotropy of the momentum-space Berry curvature. Previous studies [3–5, 9] detected a finite AHC for two perpendicular orientations of magnetic field. The magnitude of $\sigma_{yz}^A(B\|x)$ and $\sigma_{xz}^A(B\|y)$ was found to be close to each other in both Mn$_3$Ge [5] and in Mn$_3$Sn [9]. Measuring numerous samples (see Appendix), we also found that the anisotropy is small and below 0.15 (see Fig. 3a). An even smaller upper bound ($\sim$0.05) on any in-plane anisotropy is implied by our angle-dependent experiment. Our density functional theory calculations find a band structure (see the Appendix) and a Fermi surface unchanged as the spins rotate (Figs. 3b-e), in agreement with the experimental observation that finds unchanged $\sigma_{ij}^A$ for any arbitrary orientation of magnetic field in the $xy$ plane when the current is along the $z$ axis and the electric field is measured perpendicular to the magnetic field.

A finite $\sigma_{ij}^A$ arises when the overall integration of Berry curvature, $\Omega^k$ in the entire Brillouin zone does not vanish:

$$\sigma_{ij}^A = \frac{-e^2}{\hbar} \sum_n \int_{BZ} \frac{d^3k}{(2\pi)^3} f_n(k) \Omega_n^k(k). \tag{1}$$

The indexes indices $i$, $j$ and $k$ refer to the three perpendicular orientations, which are often assimilated to the $x$, $y$ and $z$ axes of the crystal lattice. Theoretical calculations [8, 11, 26–28] find Weyl nodes of opposite chirality in the vicinity of certain high-symmetry points in the Brillouin zone of Mn$_3$X materials. Because of the symmetry considerations, a finite $\sigma_{ij}^A$ is expected along certain orientations. Our result implies that this orientation is not locked to the crystal axes. The magnitude of AHC does not depend on the angle between the spin lattice and the underlying crystal. Given the geometry of the Fermi surface in the hexagonal plane (see Fig. 3b,c), this may be accounted for by assuming that the $k$-space Berry curvature reside at the vertices of the $k_z = 0$ hexagonal cut of the Brillouin zone hosting a small circular Fermi surface. A recent suggestion for the locus of the Weyl nodes [28] puts them close to these vertices, which as one sees in Figs. 3c,d host small circular sections of the Fermi surface.

## 4.3 Energy cost of domain nucleation

We now turn our attention to domain nucleation at the onset of regime II. The hysteresis loop of magnetization and Hall resistivity are shown in Figs. 4a,b. A threshold field $B_0$ of almost identical magnitude can be identified in both. Above this field, domains with a magnetization corresponding to the orientation of the applied field nucleate in the single-domain matrix that occupies the whole sample below $B_0$. As the field is swept further, the minority domain grows in size and ends up entirely replacing the former majority domain. The smooth and reproducible functional form is reminiscent of the Langevin function. However, the AHE signal increases faster than the magnetization (Fig. 4c).

The hysteresis loops were followed down to 50 K, below which the magnetic order is replaced with a spin-glass order [3]. In the whole temperature window, one could detect a finite $B_0$. Multiplying it by the jump in magnetization $\Delta M$, one quantifies the energy cost per volume

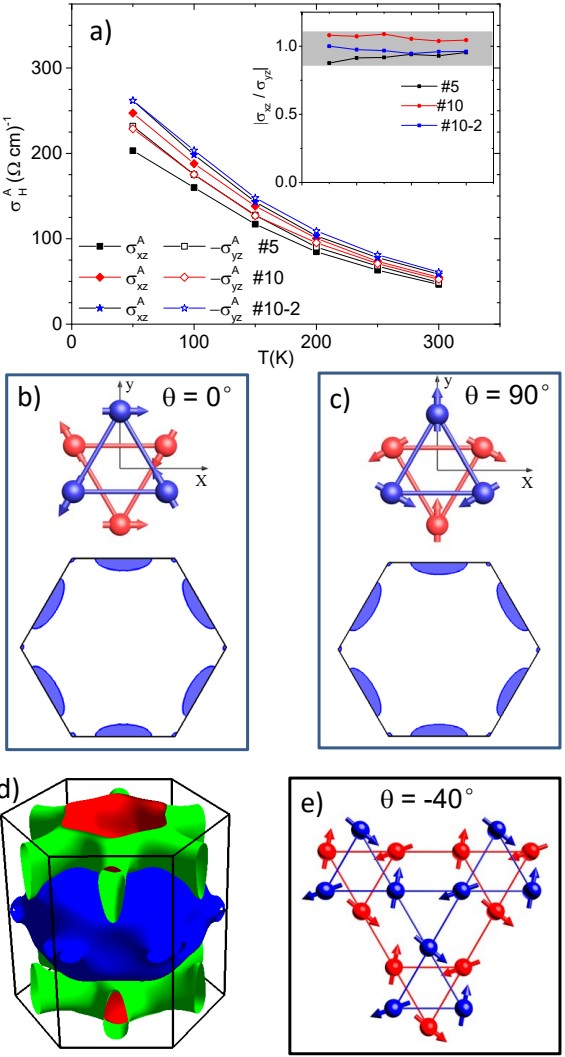

Figure 3: **In-plane isotropy:** a) Temperature-dependence of the anomalous Hall conductivity for two perpendicular orientations of the magnetic field in three different samples. The inset shows the in-plane anisotropy found in these fixed-angle measurements and the experimental margin (in gray). b,c) Mn spins in two adjacent planes (in blue and red) together with the calculated Fermi surface for each spin configuration projected in the hexagonal $(k_x, k_y, 0)$ plane of the Brillouin zone. The Fermi surface does not visibly change with spin rotation. Note also the small circular Fermi surface at the vertices. d) The calculated Fermi surface of $Mn_3Sn$ for $\theta = 0$ in three dimensions. Different colors show different Fermi surface sheets. e) Mn spins oriented along an arbitrary orientation in three adjacent six-spin David stars, each operating as a magnetic octupole [25].

of keeping the sample single domain. The temperature dependence of $E_v = B_0 \Delta M$ is shown in Fig. 4d. According to the classical theory of nucleation, the first droplet of minority domains emerges when the volume energy saved by the emergence of this domain compensates the energy cost of building a domain wall $E_S$, which is unknown. On the other hand, the amplitude of the interaction between neighboring spins $\langle J \rangle$ allows us to estimate a lower boundary to the thickness of the domain walls by using $t = (\frac{\langle J \rangle}{B_0 \Delta M})^{1/3}$. Given that $\langle J \rangle \sim 5$ meV, which is

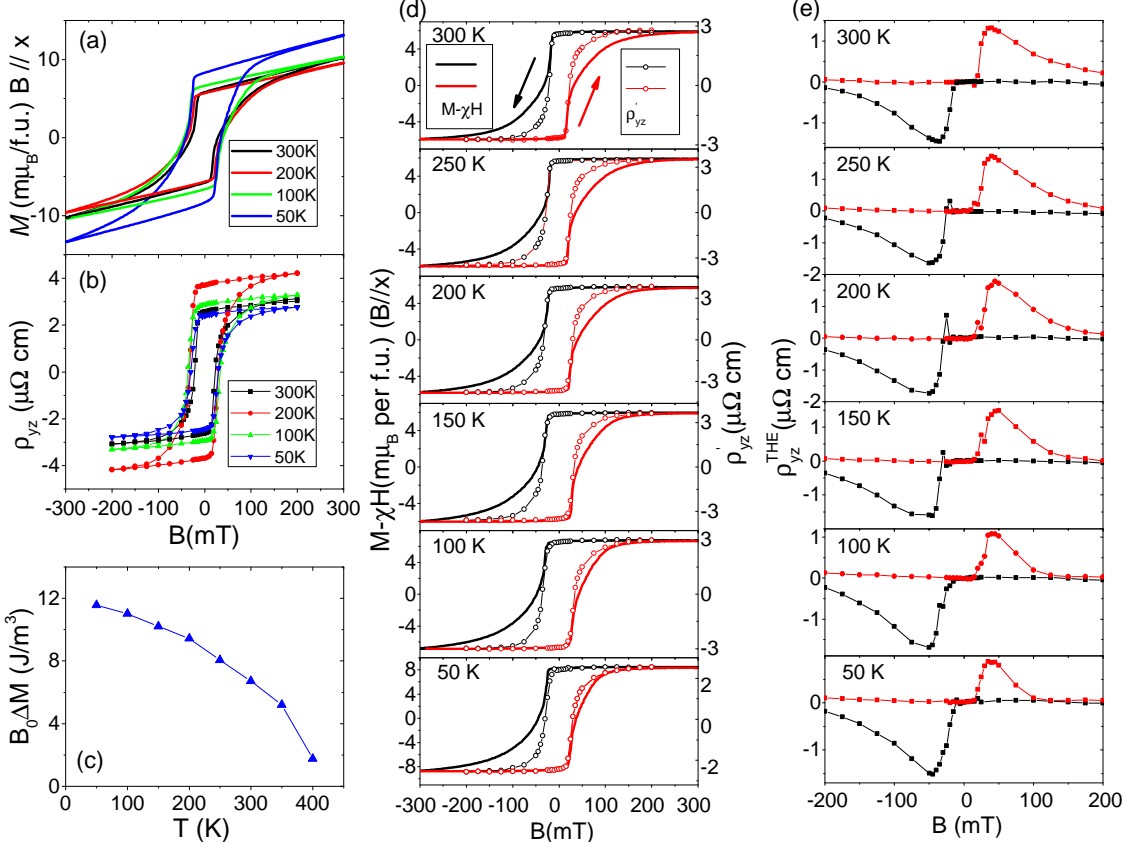

Figure 4: **Magnetization, AHE and THE:** a) Hysteretic loops of magnetization at different temperatures. b) Hysteretic loops of $\rho_{yz}$ at different temperatures. c) Temperature dependence of $B_0 \Delta M$, which represents the energy cost of staying single domain. d) Comparison of the magnetization, with its high-field slope subtracted, and anomalous Hall resistivity. The threshold field $B_0$ is identical but, at all temperatures, the magnetization shows a slower variation towards saturation at the end of the hysteresis loop. e)'Topological' Hall effect resolved by subtracting normalized magnetization times a constant from the Hall resistivity at different temperatures. At the boundary of regimes II and III, a sizeable component of the AHE is due to inhomogeneous magnetization with an out-of-plane component.

the order of magnitude of the Heisenberg coupling between spins [22] and the value of the Néel temperature, one finds $t \geq 100$ nm at room temperature. This is in agreement with what was suggested by Liu and Balents [22] by invoking the stationary solution of a sine-Gordon equation.

## 4.4 Real-space Berry curvature in the presence of domain walls

In regime II, where the system is multi-domain, such thick domain walls can be a source of Berry curvature in the real space distinct from the one provided in the momentum space by Weyl nodes. Such a distinction between components of AHE was first demonstrated in the case of MnSi. Below its Curie temperature, this helimagnet hosts a large AHE that is almost proportional to its magnetization across a wide temperature range [16] and is caused almost totally by momentum-space Berry curvature. In its A phase and in the presence of a

skyrmion lattice, an additional component to the AHE has been identified [15] and attributed to the real-space Berry curvature, which represents an effective magnetic field caused by the spatial variation of the magnetization [20, 29, 30]. In MnSi, this specific component of the AHE caused by the real-space Berry curvature [15] is an order of magnitude smaller than the total anomalous signal.

Comparing the field dependence of $\rho^A_{ij}(B)$ and $M(B)$, we observe that they do not evolve identically in regime II (See Fig. 4d). One plausible interpretation is that in the presence of domain walls, there is a distinct component of the AHE, which is not set by the magnitude of the global magnetization but is intimately linked to the presence of inhomogeneous magnetization caused by thick domain walls. In several other systems [31, 32], this has been attributed to a 'topological Hall effect' at the boundaries of a hysteresis loop. Fig. 4e shows $\rho^{THE}(B) = \rho^A_{ij}(B) - C(M(B) - B\chi))$, where $\chi$ is the high-field susceptibility (the slope of the

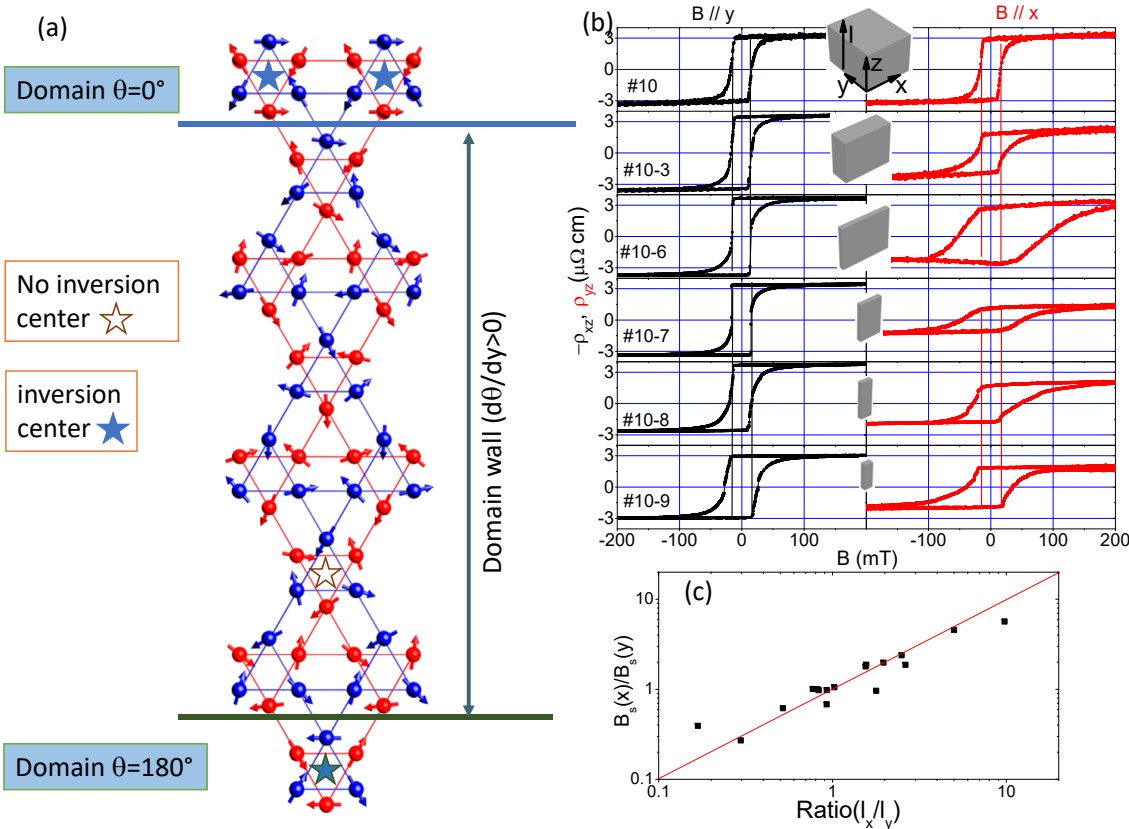

Figure 5: **Domain walls and size dependence:** a) A domain wall between two opposite domains assuming in-plane orientation for spins. Whilst in each domain, the inversion symmetry is kept, across the domain wall, it is lost due to the continuous variation of the local spin angle. This loss of inversion center provides an opportunity for the Dzyaloshinskii-Moriya interaction to generate an out-of-plane spin component, which is not shown in the present image. b) Anomalous Hall resistivity along two perpendicular orientations in a $Mn_3Sn$ single crystal after successive reductions in the sample dimensions. Note the constancy of $B_0$ and the variation of the loop width. c) The anisotropy of $B_s$, a field scale quantifying this width (see Appendix), as a function of the sample dimension ratio in a number of samples. The red solid line represents $y = x$.

magnetization outside the hysteresis loop) and $C$ is a constant. As one can see in the figure, $\rho^{THE}$ is finite in a narrow field window in regime II. We tentatively attribute this to the presence of thick domain walls, which introduce smoothly-changing magnetization that can produce a finite real-space Berry curvature [20].

However, a finite THE is expected to arise only if $\overrightarrow{n} \cdot (\frac{\partial \overrightarrow{n}}{\partial x} \times \frac{\partial \overrightarrow{n}}{\partial y})$ is finite [33]. Here, $\overrightarrow{n} = \overrightarrow{M}/M)$ is the unit vector along the orientation of magnetization [15]. If the magnetization were restricted to the plane, $\frac{\partial \overrightarrow{n}}{\partial x} \times \frac{\partial \overrightarrow{n}}{\partial y}$ would point out of the plane and the its dot product with $\overrightarrow{n}$ would be zero. Therefore, our interpretation implies the existence of a non-coplanar component in the magnetization of the domain walls. We recall that $\overrightarrow{n} \cdot (\frac{\partial \overrightarrow{n}}{\partial x} \times \frac{\partial \overrightarrow{n}}{\partial y})$ has been defined as skyrmionic number, which can be finite even in the absence of a skyrmion lattice. Our conjecture is backed by another observation. Assuming a coplanar structure, the spin configuration in the wall between two domains of opposite orientations cannot preserve the inversion symmetry (See Fig. 5a). Now, in the presence of the Dzyaloshinskii-Moriya interaction and in the absence of the inversion center, skyrmion physics is expected to emerge [34]. A very recent study of magneto-optical Kerr effect [35] confirmed that multiple domains are restricted to a narrow field window. But the internal structure of domain walls could not be resolved. Therefore, there is no direct evidence for a finite skyrmionic number in the domain walls.

We found yet another manifestation of nontrivial domain walls by detecting a correlation between the width of the hysteresis loop and sample dimensions. As seen in Fig. 5b, reducing the size of a single crystal does not modify the magnitude of $B_0$. On the other hand, it does affect the range of regime II in a remarkably intriguing way. The field-induced minority domain ends up eliminating the initial majority domain with a given rate $B_s$ (see Appendix). According to the experiment, the $B_s$ anisotropy is equal to the anisotropy of the sample dimensions parallel and perpendicular to the magnetic field (See Fig. 5c). In other words, the boundary between minority and majority domains evolves faster with increasing magnetic field along its orientation. Exploring the origin of this phenomenon and its possible connection with a bulk-edge dichotomy would be a subject matter for further theoretical and experimental studies.

# 5 Conclusion

Let us summarize the picture of the AHE in $Mn_3X$ coming out of this study. Save for a narrow field window, the system remains single-domain with a spatially homogeneous magnetization. Weyl nodes in the momentum-space are believed to be responsible for the entire AHE signal in this single-domain case. There is no trace of hexagonal symmetry of the underlying lattice in the anomalous Hall conductivity, indicating that it is entirely set by the orientation of the spin texture. We identify a narrow field window with multiple magnetic domains and therefore inhomogeneous magnetization. In this narrow window, there is a distinct contribution to the AHE which does not scale with magnetization. We suggest an interpretation for this observation by arguing that the domain walls, by possessing a non-coplanar spin component, could generate a real-space Berry curvature, which leads to an additional component of the measured AHE on top of the one produced in the momentum space. This latter conjecture shall motivate future microscopic studies of the domain walls using high-resolution magnetic imagery techniques, such as nanomagnetometery based on single nitrogen-vacancy defect in diamond [36].

## Acknowledgements

We acknowledge useful discussions with Leon Balents, Albert Fert, Jianpeng Liu, Achim Rosch, André Thiaville and Binghai Yang.

**Author contributions**   X. L., assisted by L. X. and H. Z. performed the measurements. X. L. and Z. Z. analyzed the data. Z. Z. managed the project. A. S. carried out the calculations. K. B. wrote the paper. All authors contributed to the discussion.

**Funding information**   Z. Z. was supported by the 1000 Youth Talents Plan and the work was supported by the National Science Foundation of China (Grant No. 11574097) and The National Key Research and Development Program of China (Grant No.2016YFA0401704). K. B. was supported by China High-end foreign expert programme and Fonds-ESPCI-Paris. His research was also supported in part by the National Science Foundation under Grant No. NSF PHY17-48958. Computational resources were supported by the European Research Council grant ERC-319286 QMAC and the Swiss National Supercomputing Center (CSCS) under project s575.

## A   The growth and characterization of samples

$Mn_3Sn$ single crystals were grown by the vertical Bridgman technique. For the polycrystalline samples growth, the raw materials (99.999% Mn,99.999%Sn ) were weighed and mixed inside an Ar globe box with the molar ratio of 3.3:1, and then they were put in an alumina crucible which was in a quartz ampule. The growth temperature was controlled at the bottom of the sealed vacuummed ampule. The materials were heated up to 1100 °C, remained there for 2 hour to ensure homogeneity of the melting mixture, and was cooled down slowly to 900 °C. The sample was annealed at 850 °C for 20 hours and then quenched to room temperature. For the single crystal samples growth, the polycrystalline ingot was ground and put in an alumina crucible which was in a quartz tube.Then the sealed vacuumed quartz tube was hung in a vertical Bridgman furnace. In order to get better single crystal, the single crystal sample growth was repeated three times with different rates of growth. The first growth rate is 2 mm/h and the last two growth rate is 1 mm/h. The growth temperature is 1050 °C and the growth length is 80 mm. Both the polycrystalline and single-crystalline samples were pulverized to powder for XRD measurement which confirmed the structure of $Mn_3Sn$. The single crystals were then cut into desired dimensions by a wire saw. The dimensions of some measured samples have been listed in the following tables.

## B   Computational details

Electronic structure calculations were performed within the generalized gradient approximation (GGA) of Perdew, Burke and Ernzerhof [37] using the general full-potential linearized augmented planewave method as implemented in the ELK software package [38]. Muffin-tin radii of 2.4 and 2.6 a.u. were used for Mn and Ir, respectively. The spin-orbit coupling was treated using a second-variational scheme. A $14 \times 14 \times 14$ $k$-point grid was used to perform the Brillouin zone integration, and the planewave cutoff was set by $RK_{\max} = 8$, where $K_{\max}$ is the planewave cutoff and $R$ is the smallest muffin-tin radius used in the calculations (i.e. 2.4 a.u.). The energy convergence criterion was set to 0.045 meV/Mn. Experimental lattice

parameters $a = 5.665$ and $c = 4.531$ Å and the Mn positional parameter $x = 0.8388$ were used in all our calculations [39, 40].

## C  The temperature-dependence of magnetization and Hall resistivity

We measured the temperature-dependence of magnetization and Hall resistivity using a Quantum Design PPMS and VSM. In the main text, we showed the magnetization data for the field along the $x$ axis for a selected set of temperatures. Fig. S1 shows the complete set of data along both $x$ and $y$ axes at different temperatures. In all cases, we can extract the 'topological' Hall effect (THE) also for a field along the $y$ axis as in the Fig. S2. The results for the two orientations are similar. We also measured the magnetization for a field along the $z$ axis at 300 K, as shown in Fig. S3. Our data are similar to the previous report [3]. Fig. S3 shows the Hall resistivity and magnetization for three axis at 300 K.

## D  Angle-dependent Hall resistivity in a sample with a square cross section

We also measured the angle-dependent Hall resistivity in a sample with a square cross section, in addition to the sample with a triangular cross section discussed in the main text. As shown in Fig. S4, two pairs of electrodes perpendicular to each other monitored the electric field along the $x$ ($E_x$) and $y$ axes $E_y$). The magnetic field was rotated in the basal plane while the current was applied along its $z$ axis. The total Hall resistance can be deduced from the two electric fields: $\rho_H = \sqrt{E_x^2 + E_y^2}/J_z$. The results were similar to the case of a sample with a triangle cross section. The three regimes can be clearly distinguished. In regime I, with a field lower than $B_0$, the rotation of the magnetic field does not affect the electric field: the sample remains single domain. In regime II, sweeping the angle in the basal plane back and forth produces a strong hysteresis of the Hall resistivity. In regime III, both $E_x$ and $E_y$ show sinusoidal variation and no hysteresis, indicating the spin texture of the system can be rotated easily as the magnetic field. The slight variation detected in $\rho_H$ puts an upper limit on any in-plane anisotropy.

## E  Amplitude of anomalous Hall effect in different samples

We have measured several samples to check the repeatability of our results, which are summarized in Tables S1 and S2 for 300 and 50 K, respectively. Restrictions caused by samples dimensions are the reason some measurements were not performed. As seen in the table, the magnitude of $\sigma_{yz}^A$(B∥$x$) and $\sigma_{xz}^A$(B∥$y$) were very close. Fig. S5 shows the temperature dependence of resistivity and anomalous Hall resistivity in various samples. Fig. S6 displays the temperature of anomalous Hall conductivity data for fields along the $x$ and $y$ axes compared to previous reports [3, 9].

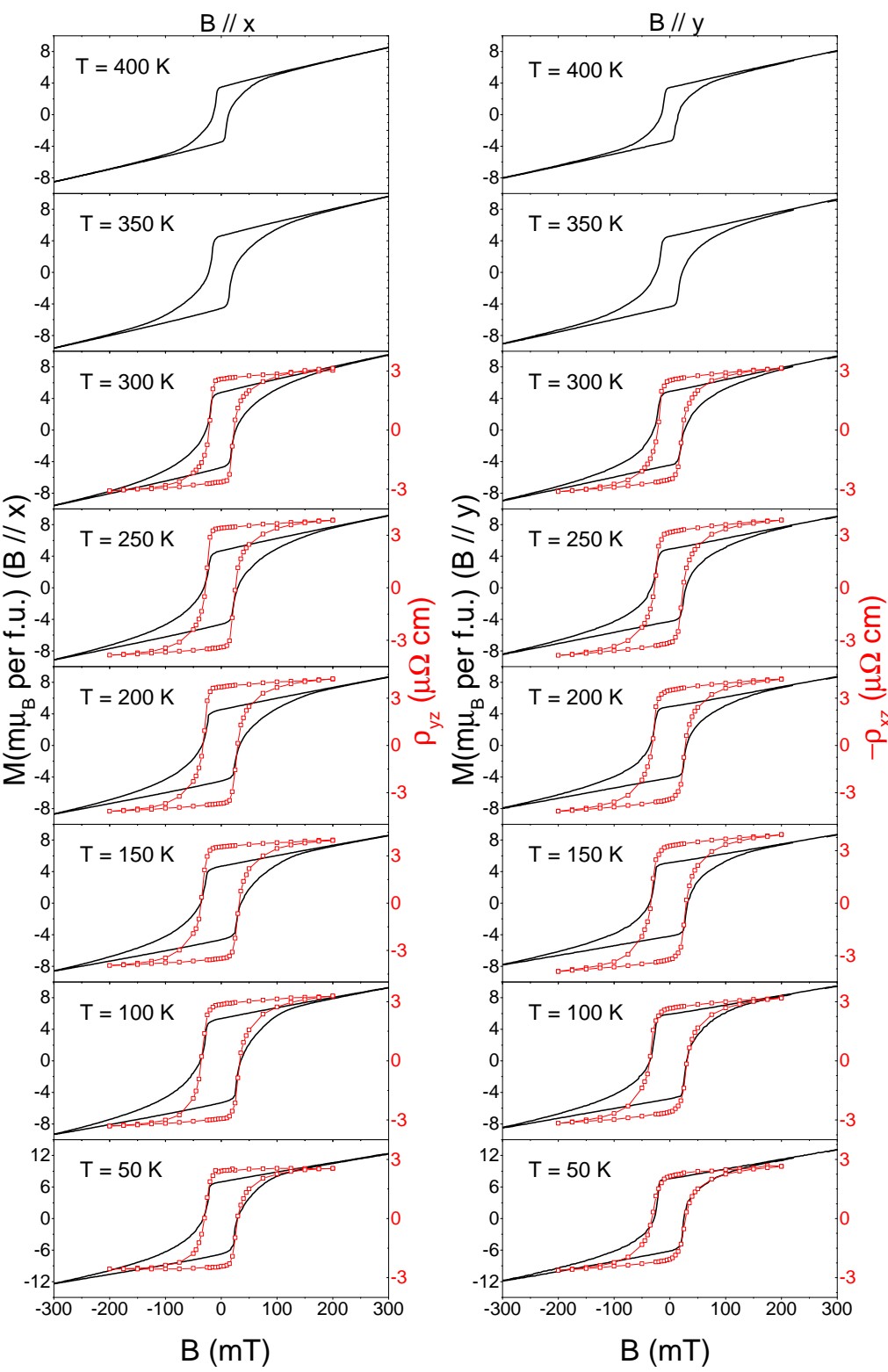

Figure S1: The magnetization and Hall resistivity with the magnetic field along $x$ (left column) and $y$ (right column) axes at different temperatures.

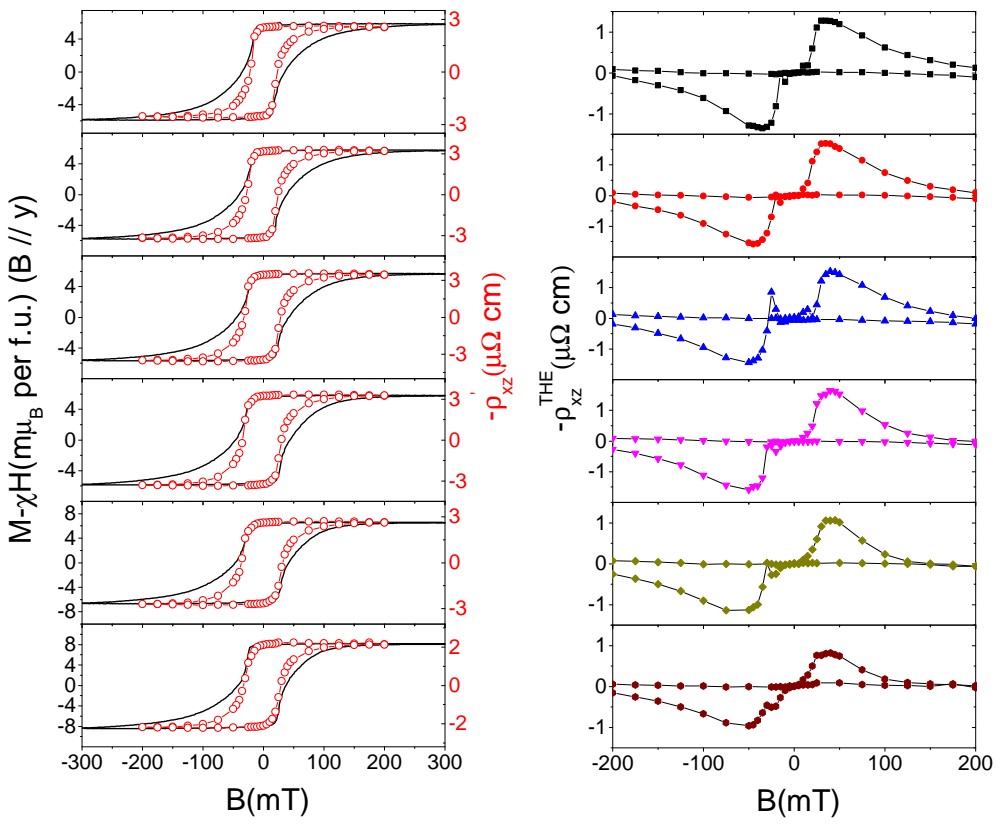

Figure S2: a) Comparison between the magnetization (with its high-field slope subtracted) and anomalous Hall resistivity for the magnetic field is along the $y$ axis. These are similar to the case with the field along the $x$ axis showed in the main text. The threshold field $B_0$ is identical, but the magnetization shows a slower variation at the end of the hysteresis loop. b) The topological Hall effect resolved by subtracting normalized magnetization times a constant from the Hall resistivity at different temperatures for a field along the $y$ axis.

## F  The extraction of $B_s$

We fitted the data between 0 to -200 mT for a positive to negative field sweep and 0 to 200 mT for negative to positive field sweep with an equation $\rho_{ij} = \rho_0(1 - 0.5e^{-B_s/B})$ to extract $B_s^+$ and $B_s^-$ respectively. $B_s$ is the average of $B_s^+$ and $B_s^-$. The Fig. S7 shows the procedure of the extraction of $B_s$ in sample # 13-2 at 300 K. $B_s$ for other samples were obtained by repeating this procedure.

## G  Comparison with MnSi

The prototype spiral helimagnet MnSi has a non-trivial magnetic texture: a skyrmion lattice in its so-called A phase. Table S3 compares the physical properties of $Mn_3Sn$ and MnSi. $S_H^A$ is defined as $\sigma_H^A/M$. For MnSi, a large AHE emerges below its Curie temperature and is almost proportional to its magnetization across a wide temperature window. The amplitude of the AHE in $Mn_3Sn$ and MnSi are comparable . On the other hand, magnetization of MnSi is

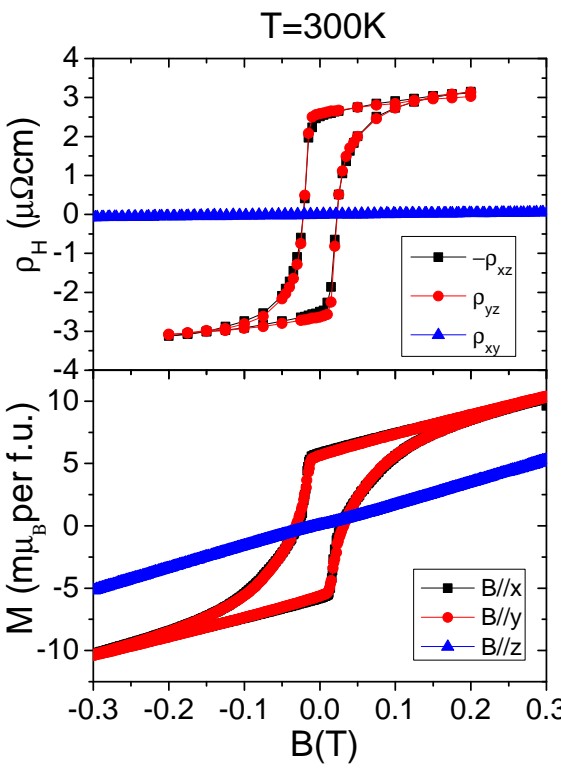

Figure S3: The magnetization and Hall resistivity for three high-symmetry axes at 300 K.

50 times larger. Therefore, the magnitude of $S_H$ (the ratio of AHC to magnetization [16]) is exceptionally large in $Mn_3Sn$ as highlighted previously [3]. Remarkably, when $Mn_3Sn$ is multidomain, a large fraction of irs AHC is caused by real-space Berry curvature. As seen in the table, this is in sharp contrast with MnSi.

## H  The amplitude of in-plane anisotropy

As discussed in the main text, we did not detect sixfold oscillations in the angle dependence of of $\sigma_{ij}^A$. Duan and co-workers [24] found a combination sixfold and twofold oscillations in torque magnetometry of of $Mn_3Sn$. They found that the angular dependence of magnetic torque $\tau$ can be described by $\tau = K_2 \sin 2\theta + \tau = K_6 \sin 6\theta$. At 270 K, they found $K_2 = 2760$ ergs/cm$^3$ and $K_6 = -2210$ ergs/cm$^3$. These numbers yield a single-ion anisotropy of the order of 0.2 $\mu$eV per Mn, in agreement with the upper boundary set by our experiment.

We also tried to calculate the in-plane magnetic anisotropy by computing the total energy as a function of the uniform spin angle rotation but did not manage to converge the total energy lower than 45 $\mu$eV per Mn with our available computing resources.

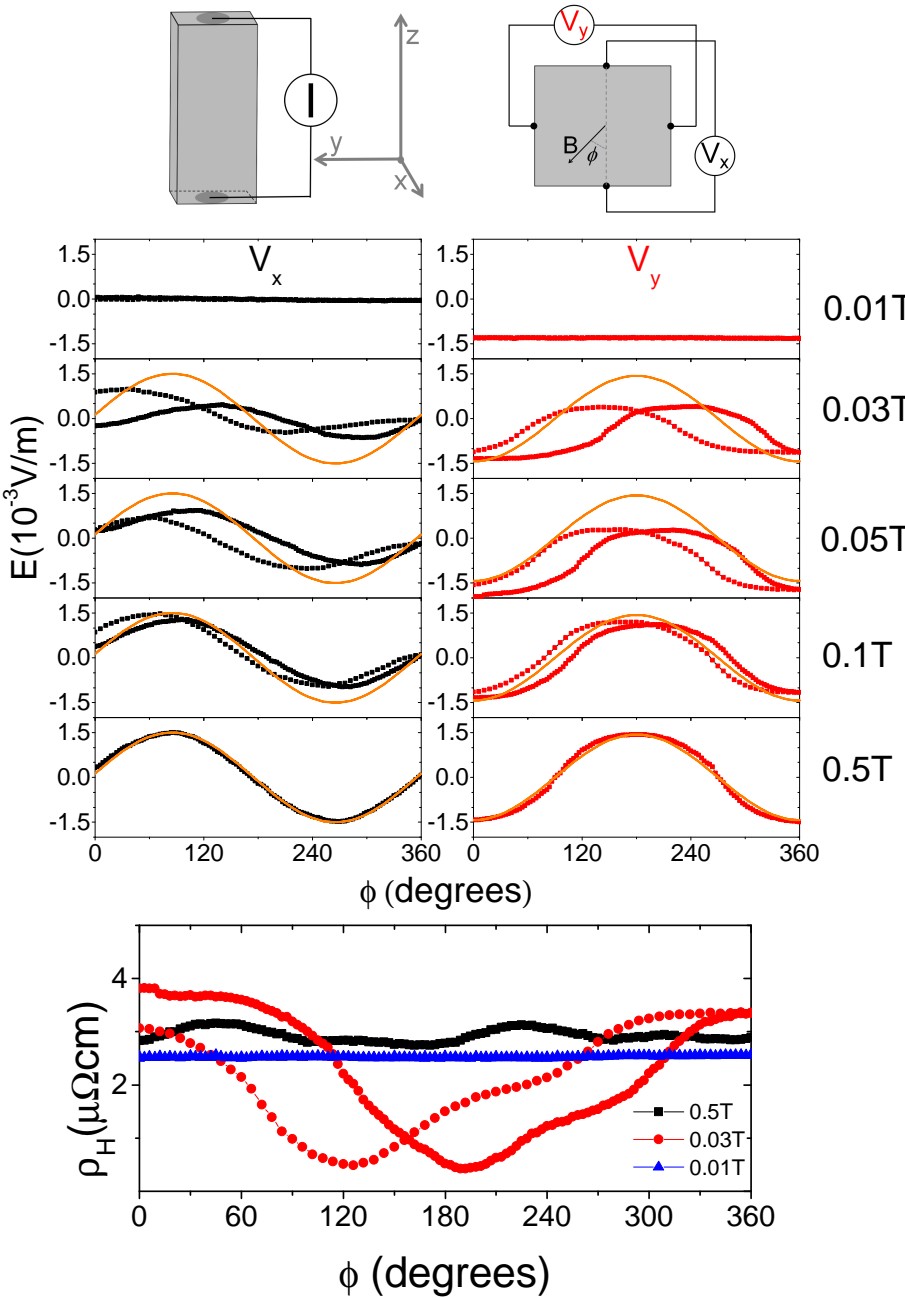

Figure S4: a) and b) The setup for measuring angle-dependent Hall resistivity in a sample with a quadrate cross-section. The current was applied along the $z$ axis and the magnetic field was rotated in the basal plane. Two pairs of electrodes monitored the electric field along the $x$ and $y$ axes. c) Angular variation of $E_x$ and d) $E_y$ as a function of the angle between the magnetic field and the $x$ axis. At low magnetic field (regime I), the total electric field remains unchanged. At high magnetic field (regime III), for both orientations, the measured electric field presents almost sinusoidal variation with almost no hysteresis. In the intermediate field range (regime II), the angular variation is strongly hysteretic. (e) The total Hall resistivity as a function of the angle.

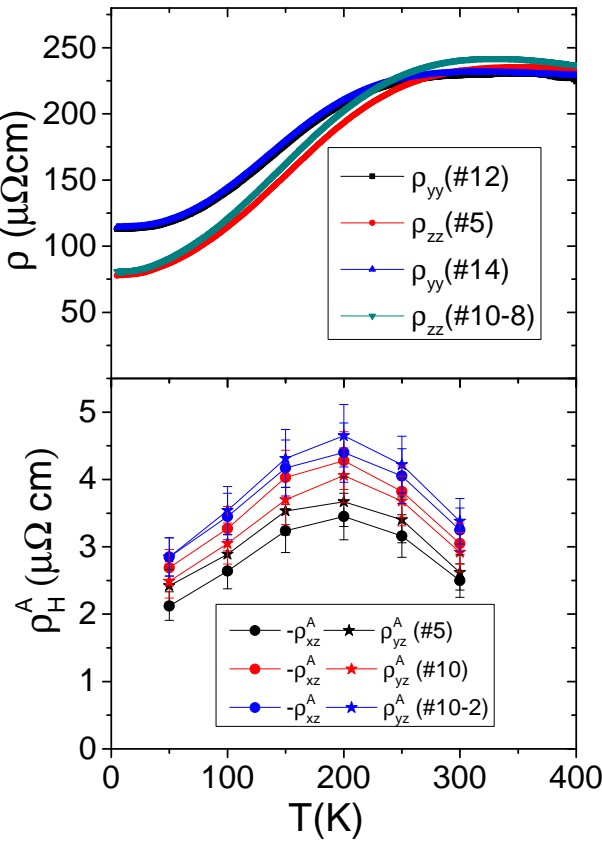

Figure S5: a) Temperature dependence of $\rho_{yy}$ and $\rho_{zz}$ in various samples. Both $\rho_{yy}$ and $\rho_{zz}$ measured in two different samples, respectively, show almost identical behavior. b) The anomalous Hall resistivity on various samples at several temperatures. Combining the data in a), we deduced the Hall conductivity of Fig. 2 in the main text.

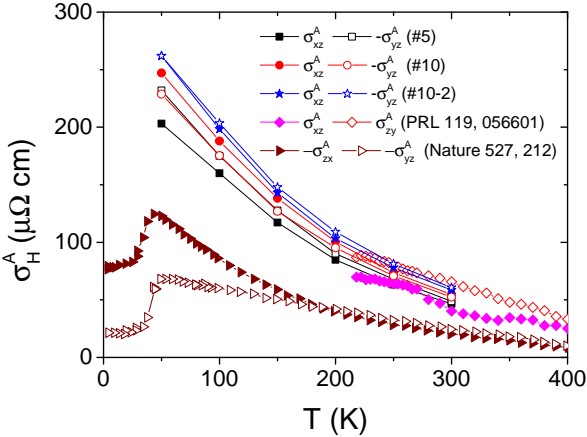

Figure S6: a) The temperature dependence of anomalous Hall conductivity in different samples and in previous reports.

Table S1: Anisotropy of the anomalous Hall effect in different samples at 300 K.

| Samples | $l_x$ mm | $l_y$ mm | $l_z$ mm | $-\rho_{xz}$ $\mu\Omega cm$ | $\rho_{yz}$ $\mu\Omega cm$ | $\rho_{xx,yy}$ $\mu\Omega cm$ | $\rho_{zz}$ $\mu\Omega cm$ | $\sigma_{xz}$ S/cm | $-\sigma_{yz}$ S/cm | -$\frac{\sigma_{xz}}{\sigma_{yz}}$ |
|---|---|---|---|---|---|---|---|---|---|---|
| #5 | 0.5 | 0.6 | 2 | 2.5 | 2.62 | | 232.5 | 46.5 | 48.7 | 0.955 |
| #6 | 0.42 | 0.38 | 1.7 | | 2.99 | | | | 53.7 | |
| #10 | 2.05 | 2.5 | 1.75 | 3.05 | 2.92 | | | 54.8 | 52.5 | 1.044 |
| #10-2 | 2.05 | 1.15 | 1.75 | 3.25 | 3.38 | | | 58.4 | 60.7 | 0.962 |
| #10-8 | 0.5 | 0.2 | 1.75 | | | | 240.6 | | | |
| #12 | 0.22 | 1.31 | 0.53 | | | 229.0 | | | | |
| #14 | 0.6 | 2.5 | 0.14 | | | 231.2 | | | | |
| Average | | | | 2.93 | 2.98 | 230.1 | 236.55 | 53.23 | 53.9 | 0.987 |

Table S2: Anisotropy of the anomalous Hall effect in different samples at 50 K.

| Samples | $l_x$ mm | $l_y$ mm | $l_z$ mm | $-\rho_{xz}$ $\mu\Omega cm$ | $\rho_{yz}$ $\mu\Omega cm$ | $\rho_{xx,yy}$ $\mu\Omega cm$ | $\rho_{zz}$ $\mu\Omega cm$ | $\sigma_{xz}$ S/cm | $-\sigma_{yz}$ S/cm | -$\frac{\sigma_{xz}}{\sigma_{yz}}$ |
|---|---|---|---|---|---|---|---|---|---|---|
| #5 | 0.5 | 0.6 | 2 | 2.12 | 2.42 | | 87.1 | 203.2 | 231.9 | 0.876 |
| #6 | 0.42 | 0.38 | 1.7 | | 2.95 | | | | 271.0 | |
| #10 | 2.05 | 2.5 | 1.75 | 2.69 | 2.49 | | | 247.1 | 228.8 | 1.08 |
| #10-2 | 2.05 | 1.15 | 1.75 | 2.85 | 2.85 | | | 261.8 | 261.8 | 1 |
| #10-8 | 0.5 | 0.2 | 1.75 | | | | 90.8 | | | |
| #12 | 0.22 | 1.31 | 0.53 | | | 118.2 | | | | |
| #14 | 0.6 | 2.5 | 0.14 | | | 119.8 | | | | |
| Average | | | | 2.55 | 2.68 | 119 | 88.95 | 237.4 | 248.4 | 0.985 |

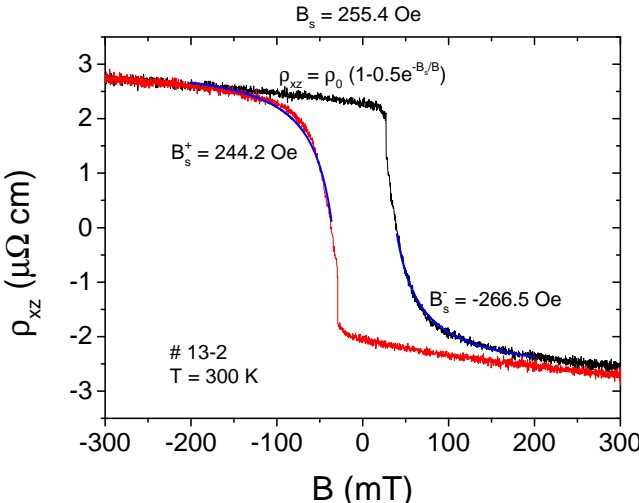

Figure S7: An example of the extraction of $B_s$ in the sample # 13-2 at 300 K

Table S3: A comparison of magnetization and AHE in $Mn_3Sn$ (at 50K) with MnSi (at 28 K).

| | $\sigma_H^A$ S/cm | $\sigma_H^{THE}$ S/cm | M A/cm | $S_H$ $V^{-1}$ | $\frac{\sigma_H^{THE}}{\sigma_H^A}$ |
|---|---|---|---|---|---|
| MnSi | 56 [15] | 1.8 [15] | 293.8 [16] | 0.19 | 0.032 |
| $Mn_3Sn$ | 232 | 113.7 | 10.75 | 21.6 | 0.49 |

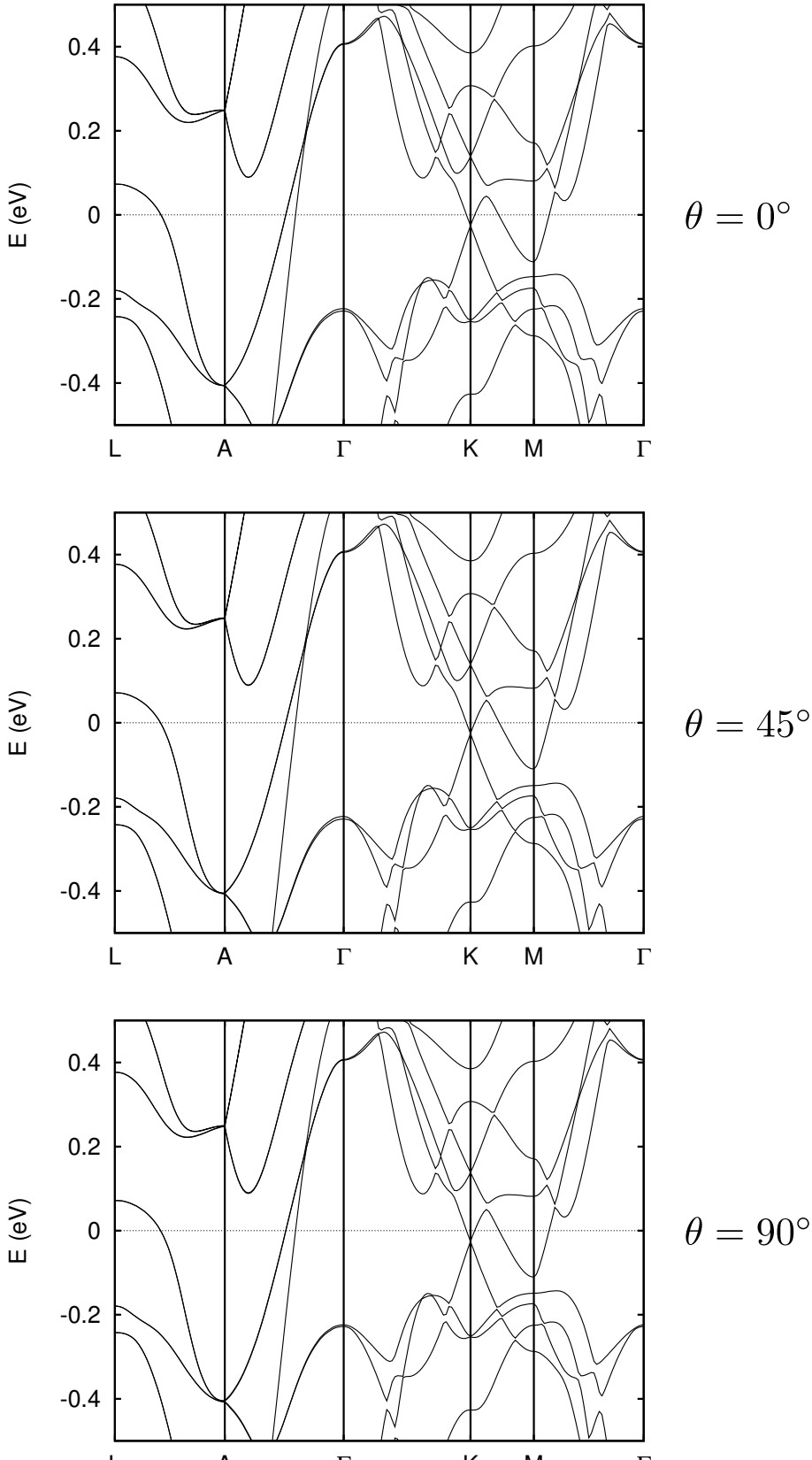

Figure S8: Band structure for three different spin orientations.

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
