# Peer review of "Momentum-space and real-space Berry curvatures in Mn$_{3}$Sn"

_SciPost Physics, doi:SciPost Phys. 5, 063 (2018)_

## Round 3 · Referee Report · Anonymous (Referee 1) · 2018-10-9

Strengths
2 - Three distinct regimes of transport are clearly identified, with hysteresis largely restricted to the middle regime.
3 - A plausible association of this phenomena with domains is established
4 - A further more subtle effect is identified and a possible origin in non-coplanar spin textures is given
5 - The clear *absence* of any evident 6-fold crystalline anisotropy is demonstrated
Weaknesses
2 - There are significant grammatical errors throughout, including even in the abstract and including such a careless error as lack of capitalization of one of the sentences. Authors should proofread at least enough to catch this type of mistake, and find a native speaker to help with this if necessary.
3 - Technically one should probably write $C_6$ rather than $Z_6$ anisotropy and domains.
4 - It is stated that the lack of dependence of the data upon sweep rate means that the system is in equilibrium (page 3). However the mere existence of two distinct states, i.e. hysteresis, means that it cannot be in equilibrium.
Report
In my opinion, the wealth of data and the interesting analysis more than justifies publication. However, the weaknesses identified above - specifically the incompatible of complete absence of anisotropy and the existence of domains, and the wrong statement about equilibrium - should be fixed.
Requested changes
1 - Statements such as "the U(1) symmetry is not broken to a Z6 anisotropy." (last sentence of Sec.4.1) should be fixed. The U(1) symmetry is definitely broken, which is unequivocal given the existence of hysteresis which can be swept by field. How it is broken is not clear from the data.
2 - Fix the wrong statement about equilibrium
3 - Carefully proofread and remove trivial grammatical mistakes, at the very least in the abstract!

Author: Kamran Behnia on 2018-10-12 [id 328]
(in reply to Report 1 on 2018-10-09)Thanks to the anonymous referee for these insightful comments, which will help us to improve our paper.
I have two quick comments:
On equilibrium- Its tricky nature is summarized by Feynman’s definition of it: “when all the fast things have happened but the slow things have not. “ According to our experiment, no matter how fast or how slow we sweep the magnetic field, the measured magnetization remains the same. Because of the hysteresis we should add to this statement “… provided that the magnetic field is swept either upward or downward. “ Recall also that in a first-order transition, hysteresis can be observed when the two states are in equilibrium (See Agarwal and Shenoy Phys. Rev. A 23, 2719( 1981)). One replace “in equilibrium” with “independent of the sweeping rate”, but I think it would be best to clarify the choice and the limitations of the word.
On domains- The referee writes: “Domains and domain walls are present *only* when the symmetry which is broken is discrete.” This is correct. However, what breaks the symmetry here is the magnetic field and not the single-ion anisotropy. The broken symmetry is indeed a discrete C6 symmetry.
Liu and Balents wrote in Phys. Rev. Lett. 119,087202 (2017): “Heisenberg exchange and Dzyaloshinskii-Moriya (DM) interaction select an approximately 120° pattern of spins with negative vector chirality, which leaves a U(1) degeneracy: any rotation of spins within the ab plane leaves the energy unchanged, when the single-ion anisotropy is neglected.”
This statement is our starting point. The application of in-plane magnetic field adds a new twist. Our experiment finds that this single-ion anisotropy is vanishingly small and what lifts the U(1) degeneracy is the in-plane magnetic field. Therefore, instead of having SIX domains (which would have been the case if the degeneracy was lifted by the single-ion anisotropy), we have only TWO domains set by the orientation of the magnetic field.
Anonymous on 2018-10-15 [id 329]
(in reply to Kamran Behnia on 2018-10-12 [id 328])This is a response from the earlier referee -- I do not quite understand the protocol for scipost. Anyway, I appreciate the clear response. I do not completely agree with the last point on domains, however. If the only term in the Hamiltonian which breaks U(1) symmetry is the magnetic field, then there will actually just be one stable or even metastable domain, not two. To get domains in the presence of a field, one needs some competing anisotropy to work against the field to oppose the force on the oppositely aligned domain (with magnetization anti-aligned to the field). To get two metastable domains, one solution would be to have a two-fold anisotropy, e.g. uniaxial within the plane. This can balance the field and yield two domains when the field is small enough. Basically one has an energy
\[ E = - a \cos2\theta - h \cos (\theta-\phi), \]
where $a$ is the two-fold anisotropy and $h$ is the Zeeman energy of the field, which is at an angle $\phi$ in the plane. When $h$ is not too large there are two local minima of this energy. This is not quite a solution because the material does seem to have 6-fold symmetry. But I do think the paper should recognize that there is some mystery here.
If not, I would be happy to see the authors offer some theory of how domains form without in-plane anisotropy and only the effect of an applied field.
Author: Kamran Behnia on 2018-10-17 [id 332]
(in reply to Anonymous Comment on 2018-10-15 [id 329])As far as I see, your argument is valid save for the fact that a finite energy cost for domain walls is not included in your equation. The attached image, is a zoom on Figure 1. Below a threshold field, the system remains single domain. It pays the energy cost of tolerating a domain with opposite magnetization to the applied field, because by this, it will avoid paying the cost of building a wall. Nucleation of a second domain (with a polarity conform to the lowest energy set by magnetic field and with a thick wall of unknown texture) starts at a field corresponding to the equality between the two energies. The one paid for bulk magnetization and the one avoided by the absence of domain walls. This is the picture invoked by classical nucleation theory in the context of hysteretic first-order phase transitions. We believe that it gives a reasonable picture of what is happening here. Admittedly, the issue of the fine structure of the interface between domains remains an open question for future theoretical and experimental studies.
I don't know much about the Scipost protocol either, but I appreciate your stimulating comments.
Attachment:

---

## Round 3 · Referee Report · Anonymous (Referee 2) · 2018-10-25

Strengths
- The topological Hall effect is revealed as a new feature of this system
- The in-plane isotropic AHE is demonstrated
- Very systematical field dependence, and insightful analysis of the transport and magnetism data
- Proposed a scenario for the THE by the possible noncoplanar spin structure in domain walls
Weaknesses
- The domain wall structure is lack of further support, for example, from the microscopic view. But the exploration of the detailed spin structure is far beyond the scope of current work.
- Some terms are misleading, such as assuming the existence of skyrmions for the THE, which is not always necessary (see below).
Report
Requested changes
-
The term of skymion may require some modification. In the discussion of the origin of THE, authors proposed the non-coplanar spin structure. Then they directly use skymions when referring to the spin structure. It is known that non-coplanar spin (or the THE) is not equivalent to the existence of skymion, although these concepts were mixed up in a lot literature. Because the THE and AHE are in the same order of magnitude, the non-coplanar spins exist in a super high density. For the "density compatible with the interatomic length scale", I prefer to call it a unique nocoplanar lattice. In other words, some spin tilting, not necessarily a skymion shape, may be enough to generate a new contribution to the AHE.
-
It is known that rho_xy = 0 for the coplanar spin structure because of the combined symmetry of Mz (z to -z) and T (time reversal symmetry). For the noncoplanar spin structure, if existing, it breaks such a combined symmetry MzT. Then it is very possible to observe a nonzero rho_xy. Therefore, I am asking whether rho_xy was measured for an in-plane (not along z) intermediate field B that induces the domain wall. If yes, it may be another strong signature.

---

## Round 4 · Referee Report · Anonymous · 2018-11-7

Report

I am satisfied with the revised manuscript and suggest the publication in current form.

---

## Round 4 · Author Response

Resubmission Comment
We are grateful to both referees for their insightful comments, which helped us to improve the manuscript. We agree with the second point raised by referee 2 that noncoplanar spin texture has implications for components of the Hall conductivity besides those discussed here. But, this is not the subject of the present already lengthy communication.

---

## Round 4 · List of Changes

Following referees’ recommendations, here are the changes in the new version:
1. We have deleted the statement "the U(1) symmetry is not broken to a Z6 anisotropy." We have detailed our discussion of single-ion-anisotropy and included a reference to an angle-dependent torque magnetometry study, in agreement with what is implied by our angle-dependent measurements of Hall conductivity.
2. We have made a more precise statement regarding the link between equilibrium and sweeping rate, by adding: “In other words, the time scale of all detectable dynamic phenomena remains faster than our sweeping rates.”
3. We have clarified the distinction and the links between ‘topological’ Hall effect, out-of-plane spin component and skyrmionic number. We specified that skyrmionic number can be finite even in absence of a skyrmion lattice.
4. We removed grammatical and spelling mistakes, which we could identify.

---

## Editorial Decision

published